# RetroTune: Mitigating spurious features via retrospective fine-tuning

## Abstract

Spurious features are non-predictive and are associated with class labels in the majority of training samples. Models trained with standard empirical risk minimization tend to base their predictions on spurious features, such as identifying objects only by their frequently co-occurring backgrounds, leading to poor performance on data without the spurious features. Mitigating a model's reliance on spurious features typically requires external supervisions, such as accurate annotations of spurious features, which are not free to get. In this paper, we propose *RetroTune*, a general self-guided spurious feature mitigation method that first inspects a model's latent representations based on the training samples for identifying unknown spurious features and then fine-tunes the model by targeting at the identified spurious features. Our method mimics the way of retrospection: it analyzes a model's latent representations for training samples after the model has been trained and then identifies and adjusts incorrect weights in the last classification layer of the model based on the analyzed results. RetroTune is fully unsupervised in identifying spurious features and does not need additional data to mitigate a model's reliance on spurious features. Our method achieves a maximum of 27.2% increment in worst-group accuracy than the best baselines on training and selecting models that are robust to *unknown* spurious features.

## 1 Introduction

Spurious features are non-predictive features in samples, such as the ocean background in images for birds recognition. A spurious feature typically exists in the majority of samples of a class but does not exist in all of them. For example, in the Waterbirds dataset (Sagawa et al., 2019), the water background feature occurs in the 95% of the waterbird images, while the remaining 5% have land backgrounds (Figure 1). In contrast, a core feature belongs to the definition of a class and always exists in class samples. Recent findings (Beery et al., 2018; Geirhos et al., 2019; 2020; Xiao et al., 2021) suggest that models trained with empirical risk minimization (ERM) tend to make confident predictions based on spurious features without the existence of core features, such as identifying an object only by its frequently co-occurring background (Geirhos et al., 2020). The reliance on spurious features can severely degrade the model's performance on the data where spurious features do not exist, posing a great challenge towards robust model generalization.

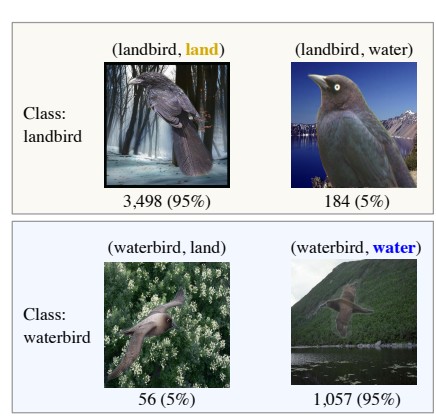

Figure 1: Example images from the Waterbirds dataset (Sagawa et al., 2019). `Land` and `water` backgrounds are the spurious features for `landbird` and `waterbird`, respectively.

Mitigating the reliance on spurious features strongly hinges on the availability of accurate annotations of spurious features, or more specifically, group labels for all (Sagawa et al., 2019) or partial (Nam et al., 2020; Liu et al., 2021; Kirichenko et al., 2022; Nam et al., 2022) training samples. A group label (`class label, spurious feature`) indicates which spurious feature is associated with which class, and a dataset can be partitioned into several groups by group labels (e.g., the Waterbirds dataset (Sagawa et al., 2019) in Figure 1). Some

models may achieve over 90% accuracy on average but degrade to random performance on certain groups of data, an indication of high dependency on spurious features. Targeting a model's worst-group performance balances optimization across groups and breaks correlations between spurious features and class labels. However, obtaining accurate group labels is not free, often requiring expensive human annotations.

In this paper, we aim for a general spurious feature mitigation method that works in most of the learning settings where group labels or other external supervisions are typically not available for both model training and selection. Our method is motivated by a recent finding (Kirichenko et al., 2022) that a model's reliance on spurious features can be attributed to the incorrect weights in the final classification layer of the model. However, retraining the last layer with a set of group-balanced data is typically infeasible as group labels are not available in most of the learning settings. Fortunately, we find that the model's latent representations, which are multi-dimensional vectors extracted before the final classification layer, provide sufficient information for us to directly identify which dimensions of latent representations are weighted incorrectly and allow us to adjust the weights in the final classification layer accordingly.

Our method, named *RetroTune*, a general self-guided approach that mimics the way of retrospection: it goes back and analyzes a model's latent representations for the training samples *after* the model has been trained to identify what dimensions of latent representations are weighted incorrectly, and fine-tunes the final classification layer targeting at the identified dimensions without requiring new training data. In particular, RetroTune exploits the number of extreme values that go beyond a significant percentile of their population distribution (Section 3.3) at each dimension of the latent representations. We show that this number can well indicate whether the corresponding dimension represents a spurious or a core feature in the original input space. Hence, we can identify dimensions that are incorrectly weighted by the classification layer by checking whether core dimensions that represents core features have higher weights than spurious dimensions that represents spurious features. We synthesize fine-tuning data by specifically perturbing at the identified dimensions of latent representations to enforce desired changes in the last layer's weights.

RetroTune is fully unsupervised in identifying spurious features and does not need additional data to mitigate a model's reliance on spurious features. Hence, it has the potential to be seamlessly integrated into many learning settings. We show that the spurious dimensions of latent representations represent non-predictive areas of the inputs and can be used to discover known spurious features with high accuracy. Our method is better than baseline methods on training and selecting models that are robust to unknown spurious features, with a maximum of 27.2% increment in worst-group accuracy than the best baselines on benchmark datasets.

## 2 RELATED WORK

**Spurious feature detection.** Spurious features are biases in data and could be harmful to a model's generalization. Discovering spurious features typically requires domain knowledge (Clark et al., 2019; Nauta et al., 2021) and human annotations (Nushi et al., 2018; Zhang et al., 2018b). Previous works discover that object backgrounds (Xiao et al., 2021) and image texture (Geirhos et al., 2019) could be spurious features that bias the predictions of deep learning models. See Geirhos et al. (2020) for an overview. Recent works (Plumb et al., 2022; Abid et al., 2022) use model explanation methods to detect spurious features. Neurons in the penultimate layer of a robust model are also used for spurious feature detection with limited human supervision (Singla & Feizi, 2022; Neuhaus et al., 2022). Our work also exploits this level of representations but does not aim for human interpretable spurious features. Moreover, our method is completely unsupervised with the ultimate goal of learning a model that is robust to spurious features. Wu et al. (2023) leverages a pre-defined concept bank as an auxiliary knowledge base for spurious feature detection. In contrast, our method does not rely on any external supervision to detect spurious features. Instead, we exploit the statistics inherent in the latent representation space of a trained model for spurious feature detection.

**Mitigating spurious bias.** Existing methods on mitigating a model's reliance on spurious features have two major assumptions on the availability of group labels. *When group labels are known*, balancing the size of the groups (Cui et al., 2019; He & Garcia, 2009), upweighting groups that do not have specified spurious correlations (Byrd & Lipton, 2019), or optimizing the worst-group performance (Sagawa et al., 2019) can be effective. *When group labels are absent*, several works

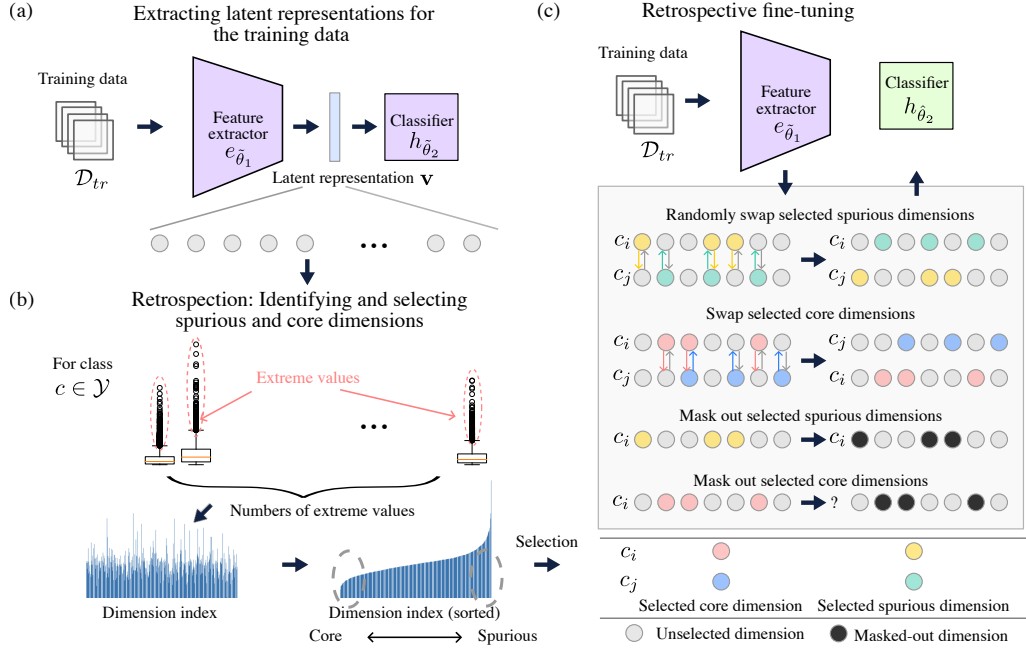

Figure 2: Overview of RetroTune for detecting and mitigating spurious features.

aim to find approximate group labels, including identifying misclassified samples (Liu et al., 2021), clustering hidden representations (Zhang et al., 2022), and invariant learning (Creager et al., 2021). Nam et al. (2022) trains a group label estimator with a part of validation data with group labels. Kirichenko et al. (2022) uses a part of balanced validation data to retrain the last layer of a model. All the methods require group labels for the validation data for model selection, which still requires non-trivial annotation efforts for every new learning task. A recent work (Asgari et al., 2022) does not require group labels and uses masked data to mitigate the impact of spurious features. However, it depends on model explanation techniques and is difficult to do model selection in practice. We focus on a general method that works in most of the learning settings where group labels are typically not available. Similar to Kirichenko et al. (2022), our method also fine-tunes the last classification layer, but we do not need a balanced validation set. Another line of works is to use data augmentation, such as mixup (Zhang et al., 2018a; Han et al., 2022; Wu et al., 2023) or selective augmentation (Yao et al., 2022), to mitigate spurious bias in model training. Our method is orthogonal to these approaches as we focus on exploiting existing representations for both spurious feature detection and mitigation.

## 3 RETROSPECTIVE FINE-TUNING

### 3.1 PROBLEM SETTING

We consider a standard classification problem in which we assume that the dataset $\mathcal{D} = \{(x, y) | x \in \mathcal{X}, y \in \mathcal{Y}\}$ contains spurious features $\mathcal{S}$ and that $\mathcal{D}$ can be partitioned into groups $G \in \mathcal{G}$, where $x$ denotes a sample in the input space $\mathcal{X}$, $y$ is the corresponding label in the finite label space $\mathcal{Y}$ with $C$ labels, $G := (y, s)$ denotes the group label defined by the combination of a class label $y$ and a spurious feature $s \in \mathcal{S}$, and $\mathcal{G}$ denotes all the group labels in the dataset. In our setting, the group information is not available during model training and selection; therefore, the learning setting is identical to the standard setting for training an ERM model. However, the goal of the problem we consider is different. Instead of optimizing the average classification accuracy, we target at training a model robust to unknown spurious features with which directly optimizing the model's worst-group accuracy is infeasible.

## 3.2 OVERALL METHOD

Our self-guided spurious mitigation method mimics the way of retrospection and consists of three major steps: (1) training an ERM model to extract its latent representations for all the training samples (Figure 2(a)), (2) a retrospection step that identifies spurious and core dimensions in latent representations (Figure 2(b)), and (3) fine-tuning the last classification layer with the synthesized training data targeted at the identified dimensions (Figure 2(c)).

**Extracting latent representations with ERM training.** We train a model $f_\theta$ with parameters $\theta$ for extracting latent representations. We use the training split $\mathcal{D}_{tr}$ of the dataset $\mathcal{D}$ to train the ERM model $f_{\tilde{\theta}}$ as follows

$$\tilde{\theta} = \arg\min_\theta \mathbb{E}_{(x,y)\in\mathcal{D}_{tr}} \ell(x, y; \theta), \tag{1}$$

where $\ell$ denotes the cross-entropy loss function. Consider the model $f_{\tilde{\theta}}$ as a feature extractor $e_{\tilde{\theta}_1}$ : $\mathcal{X} \to \mathbb{R}^D$ followed by a classifier $h_{\tilde{\theta}_2} : \mathbb{R}^D \to \mathcal{Y}$, where $D$ is the number of dimensions of latent representations for the inputs in $\mathcal{X}$, and $\tilde{\theta} = \{\tilde{\theta}_1, \tilde{\theta}_2\}$. Here, $h_{\tilde{\theta}_2}$ is the last linear layer of the model, $\tilde{\theta}_2 = \{\mathbf{W}, \mathbf{b}\}$, where $\mathbf{W} \in \mathbb{R}^{C\times D}$ and $\mathbf{b} \in \mathbb{R}^D$ denote the weight matrix and bias vector, respectively, and $e_{\tilde{\theta}_1}$ represents the remaining layers. Via the feature extractor $e_{\tilde{\theta}_1}$, we obtain the latent representation for each sample $x$ as $\mathbf{v} = e_{\tilde{\theta}_1}(x)$ in a $D$-dimensional latent space. Since the classifier $h_{\tilde{\theta}_2}$ assigns varied and fixed weights to the dimensions of $\mathbf{v}$ for each class, different dimensions of $\mathbf{v}$ represent certain fixed features related to different classes. Therefore, we consider each dimension of $\mathbf{v}$ as representing a feature of the input in the $D$-dimensional latent space. We call a dimension of $\mathbf{v}$ *spurious dimension* if it represents a spurious feature in the input space and *core dimension* if it represents a core feature.

**Retrospection: Identifying and selecting spurious and core dimensions.** We look back at the latent representations for all the training samples and detect spurious or core features by identifying spurious or core dimensions. By definition, since a core feature exists in all the training samples, values at the corresponding dimension tend to follow the same distribution. In contrast, a spurious feature does not exist in all the training samples, and values at the corresponding dimension tend to be out-of-distribution for the inputs that do not contain the spurious feature. Motivated by this, we distinguish between a core and a spurious dimension by the number of extreme values the dimension have among all the training samples. Then, we use the same information to *select* spurious and core dimensions for the following fine-tuning. See Section 3.3 for details.

**Retrospective fine-tuning.** We fine-tune the last classification layer of the model so that it assign weights that mitigate the reliance of the model on spurious features. We synthesize fine-tuning data $\mathcal{D}_{ft}$ from the latent representations of training samples with the inductive biases that: for any two latent representations, swapping core (spurious) dimensions should (should not) alter the predictions; masking out spurious dimensions should not alter the predictions; and masking out core dimensions should make the predictions uncertain.

Specifically, consider two latent representations $\mathbf{v}_i$ and $\mathbf{v}_j$ for the input $x_i$ and $x_j$ with labels $c_i$ and $c_i$. For clarity, we denote the selected spurious and core dimensions for $\mathbf{v}_i$ as $\mathbf{s}_i$ and $\mathbf{c}_i$, and $\mathbf{s}_j$ and $\mathbf{c}_j$ for $\mathbf{v}_j$. As demonstrated in Figure 2(c), we (1) *swap the selected spurious dimensions* between $\mathbf{v}_i$ and $\mathbf{v}_j$ as $\mathbf{v}_i[\mathbf{s}_j] = \mathbf{v}_j[\mathbf{s}_j]$ and $\mathbf{v}_j[\mathbf{s}_i] = \mathbf{v}_i[\mathbf{s}_i]$ without changing the labels; (2) *swap the selected core dimensions* between $\mathbf{v}_i$ and $\mathbf{v}_j$ as $\mathbf{v}_i[\mathbf{c}_j] = \mathbf{v}_j[\mathbf{c}_j]$ and $\mathbf{v}_j[\mathbf{c}_i] = \mathbf{v}_i[\mathbf{c}_i]$ while also swapping the labels; (3) *mask out the selected spurious dimensions* with $\mathbf{v}_i[\mathbf{s}_i] = 0$ and $\mathbf{v}_j[\mathbf{s}_j] = 0$ without changing the labels; and (4) *mask out the selected core dimensions* with $\mathbf{v}_i[\mathbf{c}_i] = 0$ and $\mathbf{v}_j[\mathbf{c}_j] = 0$ while making the predictions uncertain (the question mark in Figure 2(c)), where $\mathbf{v}[\mathbf{s}]$ denotes selecting elements from $\mathbf{v}$ based on the indexes in $\mathbf{s}$.

Denote $\psi : \mathbb{R}^D \times \mathcal{Y} \to \mathbb{R}^D \times \mathcal{Y}$ as a stochastic function that takes one of the four operations above with equal chance. We generate the fine-tuning data as

$$\mathcal{D}_{ft} = \{\psi(\mathbf{v}, y) | (x, y) \in \mathcal{D}_{tr}, \mathbf{v} = e_{\tilde{\theta}_1}(x)\}. \tag{2}$$

Then, we fix the feature extractor and fine-tune the classifier $h_{\theta_2}$ with $\theta_2$ initialized as $\tilde{\theta}_2$ as follows,

$$\hat{\theta}_2 = \arg\min_{\theta_2} \mathbb{E}_{(\mathbf{v},y)\in\mathcal{D}_{ft}} \ell(\mathbf{v}, y; \theta_2) + \lambda R(\mathbf{v}, y; \theta_2), \tag{3}$$

where $\ell$ denotes the cross-entropy loss function, $R(\mathbf{v}, y; \theta_2)$ denotes the negative entropy of the output probabilities from $h_{\theta_2}$, and $\lambda > 0$ is the regularization parameter. Maximizing the output entropy via $R(\mathbf{v}, y; \theta_2)$ in equation 3 reduces the prediction certainty; therefore, it naturally incorporates the objective in the fourth data generation operation above. Moreover, since the selected spurious and core dimensions are noisy estimates of the truth ones, outputs from the data generated by the other three approaches should also have reduced certainty.

### 3.3 RETROSPECTION: IDENTIFYING AND SELECTING SPURIOUS AND CORE DIMENSIONS

We identify and select spurious and core dimensions from the latent representations for all the training samples. Consider the values at the $d$'th dimension of all $\mathbf{v}$'s from the training samples of class $c \in \mathcal{Y}$ in $\mathcal{D}_{tr}$, i.e.,

$$\mathcal{V}_d^c = \{\mathbf{v}[d] | (x, y) \in \mathcal{D}_{tr}, \mathbf{v} = e_{\tilde{\theta}_1}(x), y = c\}, \quad d = 1, \ldots, D, \ c \in \mathcal{Y}, \tag{4}$$

where $\mathbf{v}[d]$ denotes the $d$'th element in $\mathbf{v}$. Then, we calculate the first quartile (25th percentile) $Q_1$, the third quartile (75th percentile) $Q_3$ and the inter-quartile range $\delta = Q_3 - Q_1$ from $\mathcal{V}_d^c$ and define the extreme value set as follows

$$\mathcal{O}_d^c = \{v | v \in \mathcal{V}_d^c, v < Q_1 - 1.5\delta \text{ or } v > Q_3 - 1.5\delta\}. \tag{5}$$

Using percentile to define an extreme value makes it adaptive and generalizable to different learning environments since each extreme value is relative to its own population distribution.

As shown in Figure 2(b), for each dimension $d$ of $\mathbf{v}$, we draw the distribution of $\mathcal{V}_d^c$ in the box plot and get the *extreme number* as $|\mathcal{O}_d^c|$, where $|\cdot|$ denotes the size of a set. Then, we sort all the extreme numbers $\mathbf{n}_c \in \mathbb{Z}^D$ in ascending order and get a list of sorted dimension indexes $\mathbf{h}$. The left-most elements of $\mathbf{h}$ are more likely to be core dimensions, while the right-most elements of $\mathbf{h}$ are more likely to be spurious dimensions (see Section 4.2 for an example). The above analysis is class-wise as different classes have different distributions of values in $\mathcal{V}_d^c$.

#### 3.3.1 HOW TO SELECT SPURIOUS AND CORE DIMENSIONS?

Although we know what dimensions are more likely to be core or spurious than others in the previous step, it is challenging to find a clear cut without any supervisions among the sorted $D$ dimensions separating core dimensions from spurious. In our method, such a cut is not required. Instead, we target on dimensions whose corresponding classifier weights are not correctly learned, e.g., higher weights assigned for spurious dimensions with larger extreme numbers.

To find such dimensions, we first search the top-$p$ *spurious dimensions* that are weighted highly by the classifier $h_{\tilde{\theta}_2}$ as follows,

$$p = \arg \max_{i=3,\ldots,D} SC(\mathbf{n}_c[\mathbf{h}_{-1}^{-i}], \mathbf{W}_c[\mathbf{h}_{-1}^{-i}]), \tag{6}$$

where $\mathbf{h}_{-1}^{-i}$ denotes the last $i$ elements of $\mathbf{h}$, $\mathbf{n}_c[\mathbf{h}_{-1}^{-i}]$ denotes the $i$ sorted extreme numbers at dimensions $\mathbf{h}_{-1}^{-i}$, $\mathbf{W}_c[\mathbf{h}_{-1}^{-i}]$ denotes the classifier weights at dimensions $\mathbf{h}_{-1}^{-i}$ for identifying class $c$, and $SC(\cdot, \cdot)$ denotes Spearman's rank correlation coefficient (Myers & Sirois, 2004) between two random variables, measuring the strength and direction of the *monotonic* relationship between the two variables. Intuitively, searching from the right end of $\mathbf{h}$ ensures that we find top-$p$ spurious dimensions that have the most extreme numbers, and finding the maximum correlation coefficient identifies dimensions whose classifier weights increase mostly in the ascending order of their extreme numbers. Therefore, the $p$ dimensions are very likely to be spurious and are weighted incorrectly by the classifier. The search index $i$ in equation 6 starts with 3, allowing enough samples for calculating the correlation coefficient. Similarly, we find top-$q$ *core dimensions* that have the smallest extreme numbers and whose corresponding classifier weights decrease mostly in the ascending order of their extreme numbers, i.e.,

$$q = \arg \min_{i=3,\ldots,D} SC(\mathbf{n}_c[\mathbf{h}_1^i], \mathbf{W}_c[\mathbf{h}_1^i]), \tag{7}$$

where $\mathbf{h}_1^i$ denotes the first $i$ elements of $\mathbf{h}$. The selected $q$ dimensions are mostly assigned with correct classifier weights and we aim to keep them as core dimensions during fine-tuning. In contrast, the selected $p$ dimensions are mostly assigned with incorrect classifier weights which we aim to adjust during fine-tuning.

### 3.3.2 When to extend the selected spurious dimensions?

After finding $p$ and $q$, we select $\mathbf{h}_{-1}^{-p}$ as the spurious dimensions and $\mathbf{h}_1^q$ as the core dimensions. The dimensions $\mathbf{h}_{q+1}^{D-p}$ in between are not clearly recognized by the classifier $h_{\tilde{\theta}_2}$ as core or spurious dimensions. If values at dimensions $\mathbf{h}_{q+1}^{D-p}$ have large variations, then these dimensions may affect the robustness of the classifier. Hence, we extend the selected spurious dimensions to include $\mathbf{h}_{q+1}^{D-p}$ for mitigating the potential impact from these dimensions. We develop a simple heuristic called *partition entropy* to detect this case. First, for each class $c \in \mathcal{Y}$, we calculate the sum of variances on the left and right halves of $\mathbf{h}$ as $\sigma_l = \sum_{i=1}^{D/2} var(\mathcal{V}_{\mathbf{h}[i]}^c)$ and $\sigma_r = \sum_{i=D/2}^{D} var(\mathcal{V}_{\mathbf{h}[i]}^c)$, respectively, where $\mathcal{V}_{\mathbf{h}[i]}^c$ is defined in equation 4 for the $\mathbf{h}[i]$'th dimension. After normalizing $\sigma_l$ and $\sigma_r$ with their sum, we get $\sigma_l'$ and $\sigma_r'$. If the two values are close enough with the *partition entropy* $H > 0.99$, where $H = -\sigma_l' \log \sigma_l' - \sigma_r' \log \sigma_r'$, then we say the dimensions $\mathbf{h}_{q+1}^{D-p}$ have significant variations and should be added to the selected spurious dimensions which now become $\mathbf{h}_{q+1}^D$.

### 3.4 Model selection with unknown spurious features

In training an ERM model, we select the model that achieves the best average classification accuracy $a_{avg}$ on the validation set $\mathcal{D}_{val}$. During fine-tuning, selecting a model that is robust to spurious features becomes challenging since no group labels are available. Without group labels, the worst-group classification accuracy, a metric that quantifies a model's robustness to spurious features, cannot be obtained. To address this, we first consider the set of misclassified validation samples as $\mathcal{D}_{val\_err} = \{(x,y)|f_{\tilde{\theta}}(x) \neq y, (x,y) \in \mathcal{D}_{val}\}$. Then, we calculate the accuracy $a_{err}$ on $\mathcal{D}_{val\_err}$ as a proxy for the worst-group accuracy. To further balance between worst and average performance, we consider the harmonic mean of $a_{avg}$ and $a_{err}$, i.e.,

$$a_\beta = \frac{\beta a_{avg} a_{err}}{a_{avg} + \beta a_{err}}, \tag{8}$$

where $\beta$ is a scalar adjusting the relative importance between $a_{err}$ and $a_{avg}$. A large $\beta$ favors a model with a high $a_{avg}$ but may select a model with poor worst-group accuracy. In practice, we select $\beta$ such that the difference between $a_{avg}$ on the ERM model and the fine-tuned model meets with our expectation, e.g., allowing a maximum of 5% drop in $a_{avg}$ after fine-tuning. We provide a detailed discussion in Appendix A.4 on how to choose $\beta$ for model selection.

## 4 Experiment

### 4.1 Experiment setup

**Datasets.** We test RetroTune on three image datasets with various types of spurious features. **Waterbirds** (Sagawa et al., 2019) is a dataset for recognizing waterbird and landbird. It is generated synthetically by combining images of the two birds from the CUB dataset (Welinder et al., 2010) and the backgrounds, water and land, from the Places dataset (Zhou et al., 2017), producing (waterbird, water), (waterbird, land), (landbird, land), and (landbird, water) groups. **CelebA** (Liu et al., 2015) is a large-scale image dataset of celebrity faces. The task is to identify hair color, non-blond or blond, with genders as the spurious features. There are four groups in the CelebA dataset: (blond, male), (blond, female), (non-blond, male), (non-blond, female). **Biased MNIST** (Bahng et al., 2020) introduces color biases that highly correlate with the labels during training. We use this dataset to test models with controlled spurious biases. We give detailed descriptions in Appendix A.3.

**Training settings.** We use the ResNet-50 pre-trained on ImageNet as the feature extractor. We use an SGD optimizer with an initial learning rate of 0.001, a momentum of 0.9, a weight decay of $10^{-4}$, and a cosine annealing learning rate scheduler during ERM training. We fine-tune an ERM model using Adam optimizer (Kingma & Ba, 2014) with a constant learning rate. We set $\lambda = 0.1$, $\beta = 5$ in all our experiments. We provide training details in Appendix A.2 and results on transformer-based architectures (Dosovitskiy et al.) in Appendix A.5. All experiments are conducted on NVIDIA RTX A6000 GPUs.

**Evaluation.** We assume that group labels are available in test data. We define the *worst-group classification accuracy* as the lowest accuracy achieved among all possible data groups, and the

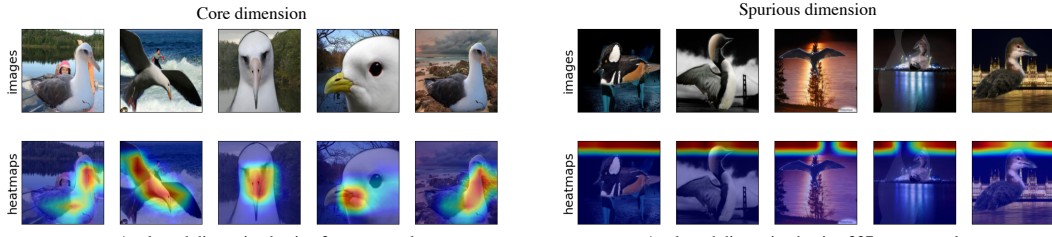

Figure 3: Visualization of the identified spurious and core dimensions from a ResNet-50 model trained on the Waterbirds dataset. We use neural activation maps (Singla & Feizi, 2022) to highlight the region related to a selected dimension.

| Top-K dimensions | 5 | 10 | 15 | 20 | 50 | 100 | 2048 (all) |
|---|---|---|---|---|---|---|---|
| ARI ↑ | 0.89 | **0.92** | 0.92 | 0.92 | 0.87 | 0.87 | 0.87 |

Table 1: Adjusted rand indexes (ARI) for inferring the group labels for the training samples in the Waterbirds dataset using different numbers of dimensions of latent representations.

*average classification accuracy* as the accuracy averaged over all test samples. The worst-group accuracy is a standard metric for measuring a model's robustness to spurious features. Since having high average accuracy does not necessarily lead to high worst-group accuracy, and having high worst-group accuracy may adversely affect average accuracy, we report both metrics. For each of our experiments, we report results averaged over five runs.

## 4.2 EXTREME NUMBER IS INDICATIVE OF SPURIOUS FEATURES

To justify our use of extreme numbers (Section 3.3) for identifying spurious features, we first visualize which part of an input a selected dimension represents. We trained a ResNet-50 model on the Waterbirds dataset using ERM. Then, for each class in the dataset, we sorted the extreme numbers in ascending order and obtained the corresponding sorted dimension indexes. We selected two dimensions from the left and right sides of the sorted dimension indexes, respectively, and generated the activation heatmaps for the two dimensions. For each dimension, we chose top-five images from the `waterbird` class that have the largest values at the dimension. Images shown in the left panel of Figure 3 are selected for the dimension having 3 extreme values, while images shown in the right panel are selected for the dimension having 227 extreme values. The dimension with very few extreme values represents a core feature as it points to areas that mostly lie within birds. In contrast, the dimension having many extreme values represents a spurious feature as it points to areas that are non-predictive of the class. More visualizations are shown in Appendix A.3.

For quantitative evaluation, we demonstrate that spurious dimensions selected by extreme numbers can be used to infer the group labels of training samples with high accuracy. A group label (Section 3.1) specifies an association between a spurious feature and a class label. To infer group labels, we cluster the latent representations of the training samples in the same class and assign cluster indexes to the samples as their spurious features. In the experiment on the Waterbirds dataset, we used K-means with 2 clusters and selected top-$K$ dimensions that have most extreme values for clustering. We calculated the adjusted rand index (ARI) (Milligan & Cooper, 1986) between inferred and true group labels, with a higher ARI indicating a better clustering. Results in Table 1 show that with just top-10 spurious dimensions or 0.488% of all dimensions, our method achieves the highest ARI. Using more dimensions becomes ineffective as it gradually include core dimensions which are not predictive of spurious features. Overall, the number of extreme values at each dimension is indicative of spurious features.

## 4.3 ABLATION STUDIES

**Analysis on extending spurious dimensions.** We extend the selected spurious dimensions based on the simple heuristic, partition entropy, proposed in Section 3.3.2. To justify this, we give an example based on the CelebA dataset. We first trained an ERM model on the dataset. Then, for each class,

| Dataset | Class | Core | Spurious | Extended spurious | Acc. (-) | Worst (-) | Acc. | Worst |
|---------|-------|------|----------|-------------------|----------|-----------|------|-------|
| Waterbirds | Landbird | 5 | 74 | 2043 | 92.9 | 87.0 | **92.9** | **87.5** |
| | Waterbird | 12 | 4 | 4 | | | | |
| Celeba | Non-blond | 8 | 7 | 2040 | **92.5** | 73.3 | 91.0 | **82.2** |
| | Blond | 3 | 20 | 20 | | | | |

Table 2: Classification accuracy (%) comparison between models trained on the fine-tuning data generated with and without (marked with "(-)") extended spurious dimensions. Numbers of selected core, spurious, and extended spurious dimensions for each class are listed for comparison.

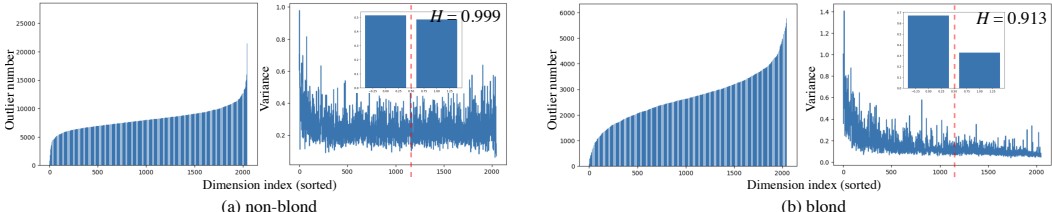

(a) non-blond                                  (b) blond

Figure 4: Extreme numbers and variances at each dimension of latent representations for the (a) `non-blond` and (b) `blond` classes obtained from a ResNet-50 model trained on the CelebA dataset. Dimension indexes are sorted in the ascending order of extreme numbers. The partition entropy along with its underlying distribution is shown for each class.

we calculated the extreme number using equation 5 and the variance for each dimension of latent representations. Most dimensions in the middle have relatively large variances in comparison to the spurious ones (right side) (Figure 4), resulting in a high partition entropy (over 0.99). Targeting at a few right-most spurious dimensions ignores those in the middle, making the classifier unrobust to changes in the middle dimensions. In contrast, for latent representations from the `blond` class, we observe relatively small variances in the middle dimensions and a smaller partition entropy in Figure 4(b). Table 2 shows that extending spurious dimensions for classes that have high partition entropies, such as the `non-blond` class in CelebA, improves worst-group accuracy.

**Effectiveness of the data generation operations.** We analyze the effectiveness of the four data generation operations using a leave-one-out strategy. Specifically, we tested models fine-tuned on the data generated by any three of the data generation operations and reported the results in Table 3. For the Waterbirds dataset, we observe that each of the four data generation operations is important since removing any of them lowers the worst-group accuracy. Moreover, operations on the selected core dimensions are more effective in improving the average and worst-group accuracies than on the selected spurious dimensions. For the CelebA dataset, the only active operation is Swap-S as we observe no more than 1% drop in average accuracy and approximately 1% increases in the worst-group accuracy on the other three operations. The varied effectiveness of the four operations on the two datasets directly relates to the numbers of selected spurious and core dimensions: the average number of selected core dimensions per class on the Waterbirds dataset is larger than that on the CelebA dataset (Table 2). Intuitively, identifying hair colors is easier than identifying birds of different kinds. Nevertheless, the proposed four data generation operations have no significant adverse effects on a model's average and worst-group performance and can be active or inactive depending the learning tasks.

## 4.4 COMPARISON WITH BASELINE METHODS

We compare our method with baseline methods for improving a model's robustness to spurious features, including LfF (Nam et al., 2020), CVaR DRO (Levy et al., 2020), JTT (Liu et al., 2021), DFR (Kirichenko et al., 2022), and DivDis (Lee et al., 2022). When there are no group labels for model training and selection, most methods suffer from spurious features, with large gaps between average and worst-group accuracies. In this challenging setting, our method still significantly improves the worst-group accuracy, with 6.5% and 27.2% higher than the best baseline DivDis on the Waterbirds and CelebA datasets, respectively.

| Dataset | Swap-S (-) | | Mask-S (-) | | Swap-C (-) | | Mask-C (-) | | All | |
|---|---|---|---|---|---|---|---|---|---|---|
| | Avg. | Worst | Avg | Worst | Avg. | Worst | Avg. | Worst | Avg. | Worst |
| Waterbirds | 92.1 | 84.0 | 93.7 | 86.0 | 89.0 | 78.1 | 92.8 | 80.9 | 92.9 | 87.5 |
| CelebA | 91.6 | 76.1 | 90.1 | 83.2 | 90.2 | 83.9 | 90.4 | 83.1 | 91.0 | 82.2 |

Table 3: Leave-one-out test for the four types of data generation operations: Swap-S for swapping between spurious dimensions, Mask-S for masking out spurious dimensions, Swap-C for swapping between core dimensions, and Mask-C for masking out core dimensions. "(-)" denotes removing the corresponding operation. "All" denotes using all the four operations. Average and worst-group accuracies (%) are reported.

| Dataset | Metric | ERM | LfF | CVaR DRO | JTT | DFR | DivDis | RetroTune (Ours) |
|---|---|---|---|---|---|---|---|---|
| Waterbirds | Avg. | 91.9 | 91.2 | 95.2 | 93.3 | 92.1 | 90.7 | 92.9 |
| | Worst | 75.7 | 44.1 | 62.0 | 62.5 | 77.4 | 81.0 | **87.5** |
| CelebA | Avg. | 96.0 | 85.1 | 82.5 | 88.0 | 95.8 | 90.8 | 91.0 |
| | Worst | 42.2 | 24.4 | 36.1 | 40.6 | 46.0 | 55.0 | **82.2** |

Table 4: Average and worst-group accuracy comparison (%) when no group labels are available for model training and selection.

## 4.5 EVALUATION WITH CONTROLLED BIASES

We further test our method in extremely biased settings with the Biased MNIST dataset (Bahng et al., 2020). The training set explicitly controls the proportion $\rho$ of a digit class having a particular color background, and a larger proportion indicates a stronger correlation between a digit class and a color background during training. Each class in the test set has balanced color backgrounds. In Table 5, we compare our method with ReBias (Bahng et al., 2020), an effective method in learning under extremely biased settings. We observe that in the extremely biased settings when $\rho \in \{0.999, 0.997\}$, ReBias has better average classification accuracies. We reason that in such settings, latent representations from an ERM model provide little information about spurious and core features since the model is not well trained. In practice, extremely biased learning environment is rare. In a less biased setting with $\rho = 0.99$, i.e., with 1% training samples having random color backgrounds, our method performs the best in both metrics.

| $\rho$ | 0.999 | | 0.997 | | 0.995 | | 0.990 | |
|---|---|---|---|---|---|---|---|---|
| | Acc. | Worst | Acc. | Worst | Acc. | Worst | Acc. | Worst |
| ERM | 10.0 | 0.0 | 45.2 | 0.0 | 76.0 | 2.2 | 90.0 | 50.0 |
| ReBias | **15.4** | 0.0 | **62.1** | **1.0** | 73.7 | **6.8** | 87.1 | 20.8 |
| RetroTune (Ours) | 10.0 | 0.0 | 45.8 | 0.0 | **76.3** | 3.3 | **90.0** | **54.1** |

Table 5: Average and worst-group accuracy (%) comparison on the Biased MNIST dataset.

## 5 CONCLUSION

We focused on a general spurious mitigation method that works in most of the learning settings where external supervisions are typically not available. We motivated our method by mimicking the way of retrospection. We analyzed a model's latent representations for all the training data after the model has been trained and showed that the extreme number at a dimension of latent representations has a positive correlation with the likelihood of the dimension representing a spurious feature. We developed an unsupervised method utilizing this statistic to identify spurious dimensions that are weighted incorrectly by the last classification layer. We proposed retrospective fine-tuning that targets at the identified dimensions to mitigate the model's reliance on spurious features. Our method, RetroTune, does not need additional data and is effective in learning a model that is robust to unknown spurious features.

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

# A  APPENDIX

## A.1  DATASETS

We give the descriptions of the Waterbirds, CelebA, and Biased MIST datasets in Figures 5, 6, and 7, respectively. The Waterbirds dataset is used to test whether a model relies on the background of an image to identify its class membership. The CelebA dataset is used to test whether a model relies on gender to predict hair color. The Biased MNIST dataset is used to test whether a model relies on background colors associated with each class for predictions. The proportions of spurious features in each class are fixed for the Waterbirds and CelebA datasets. In Biased MNIST, we can tune the proportion $\rho$ to explicitly control the correlation between a class label and a background color.

| Waterbirds dataset | | | |
|---|---|---|---|
| Group label | (landbird, land) | (landbird, water) | (waterbird, land) | (waterbird, water) |
| Images |  |  |  |  |
| Class label | landbird (0) | landbird (0) | waterbird (1) | waterbird (1) |
| #Train data | 3,498 (73%) | 184 (4%) | 56 (1%) | 1,057 (22%) |
| #Val data | 467 | 466 | 133 | 133 |
| # Test data | 2,255 | 2,255 | 642 | 642 |
| Spurious features: water and land backgrounds | | | |

Figure 5: Descriptions of the Waterbirds dataset.

| CelebA dataset (hair color classification) | | | |
|---|---|---|---|
| Group label | (non-blond, female) | (non-blond, male) | (blond, female) | (blond, male) |
| Images |  |  |  |  |
| Class label | non-blond (0) | non-blond (0) | blond (1) | blond (1) |
| #Train data | 71,629 (44%) | 66,874 (41%) | 22,880 (14%) | 1,387 (1%) |
| #Val data | 8,535 | 8,276 | 2,874 | 182 |
| # Test data | 9,767 | 7,535 | 2,480 | 180 |
| Spurious features: female and male | | | |

Figure 6: Descriptions of the CelebA dataset.

## A.2  TRAINING DETAILS

**Experiments on the Waterbirds dataset.** For training an ERM model, we use the ResNet-50 pre-trained on ImageNet as the model initialization. We use an SGD optimizer with an initial learning rate of 0.001, a weight decay of $10^{-4}$, a momentum of 0.9. We set the batch size to 64 and train the model for 300 epochs. The learning rate decay is controlled by a cosine annealing scheduler. Standard data augmentations are used during model training. For our RetroTune training, we set $\lambda$ to 0.1 and $\beta$ to 5. We first generate embeddings for all the training samples without using data

| Biased MNIST | | | | | | | | | | |
|---|---|---|---|---|---|---|---|---|---|---|
| Class label | 0 | 1 | 2 | 3 | 4 | 5 | 6 | 7 | 8 | 9 |
| Proportion $\rho$ | | | | | | | | | | |
| Fixed background colors | | | | | | | | | | |
| Proportion $1-\rho$ | | | | | | | | | | |
| Random background colors | | | | | | | | | | |

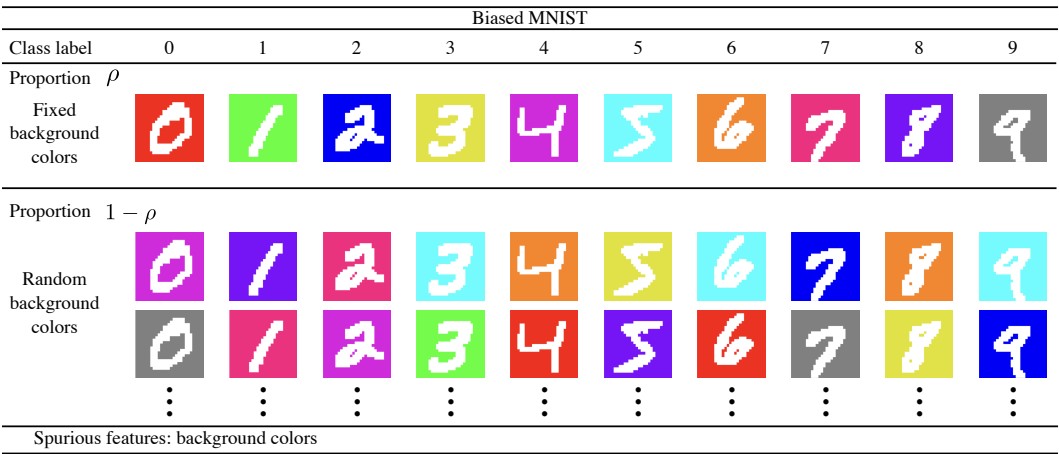

Figure 7: Descriptions of the Biased MNIST dataset.

augmentation. We use an Adam optimizer with a constant learning rate of $10^{-4}$. We set the batch size to 256 and fine-tune the last classification layer for 20 epochs.

**Experiments on the CelebA dataset.** For training an ERM model, we use the ResNet-50 pre-trained on ImageNet as the model initialization. We use an SGD optimizer with an initial learning rate of 0.001, a weight decay of $10^{-4}$, a momentum of 0.9. We set the batch size to 128 and train the model for 100 epochs. The learning rate decay is controlled by a cosine annealing scheduler. Standard data augmentations are used during model training. For our RetroTune training, we set $\lambda$ to 0.1 and $\beta$ to 5. We first generate embeddings for all the training samples without using data augmentation. We use an Adam optimizer with a constant learning rate of $10^{-5}$. We set the batch size to 256 and fine-tune the last classification layer for 50 epochs.

**Experiments on the Biased MNIST dataset.** We use a four-layer convolutional network with 16, 32, 64, 128 filers and a kernel size of 7. We use an Adam optimizer with an initial learning rate of 0.001. We set the batch size to 256 and train the model for 80 epochs. The learning rate decays with a factor of 0.1 after every 20 epochs. For our RetroTune training, we set $\lambda$ to 0.1 and $\beta$ to 5. We first generate embeddings for all the training samples. We use an Adam optimizer with a constant learning rate of $10^{-5}$. We set the batch size to 256 and fine-tune the last classification layer for 1 epoch.

### A.3 HEATMAPS OF SELECTED DIMENSIONS

We provide visualization in Figure 8 and Figure 9 for additional dimensions selected from the latent embedding space (2048 dimensions) of a ResNet-50 trained on the Waterbirds and CelebA datasets, respectively. Dimensions with many extreme values tend to represent spurious features that are non-predictive, while those with few extreme values tend to represent core features that lie within the definition of an object.

### A.4 THE MODEL SELECTION METRIC

We propose a model selection metric in equation 8 with a pre-specified parameter $\beta$. A larger $\beta$ favors a model that has a larger classification accuracy; however, the selected model tends to have lower worst-group accuracy. As shown in Figure 10, when models are trained on the CelebA dataset, a larger $\beta$ selects a model that has a larger average classification accuracy; however, the worst-group accuracy decreases after reaching a peak point. Therefore, using a large $\beta$, i.e., when $\beta > 3$, has a tradeoff between the two accuracies on the CelebA dataset. Since the worst-group accuracy curve after $\beta > 3$ is steep in comparison with the average accuracy curve, a small compromise in average performance can select a model that has significant improvement on worst-group accuracy. In contrast, on the Waterbirds dataset, regardless of the values of $\beta$, the selected models tend to have similar average and worst-group accuracies. Therefore, the selection metric is robust on the Water-

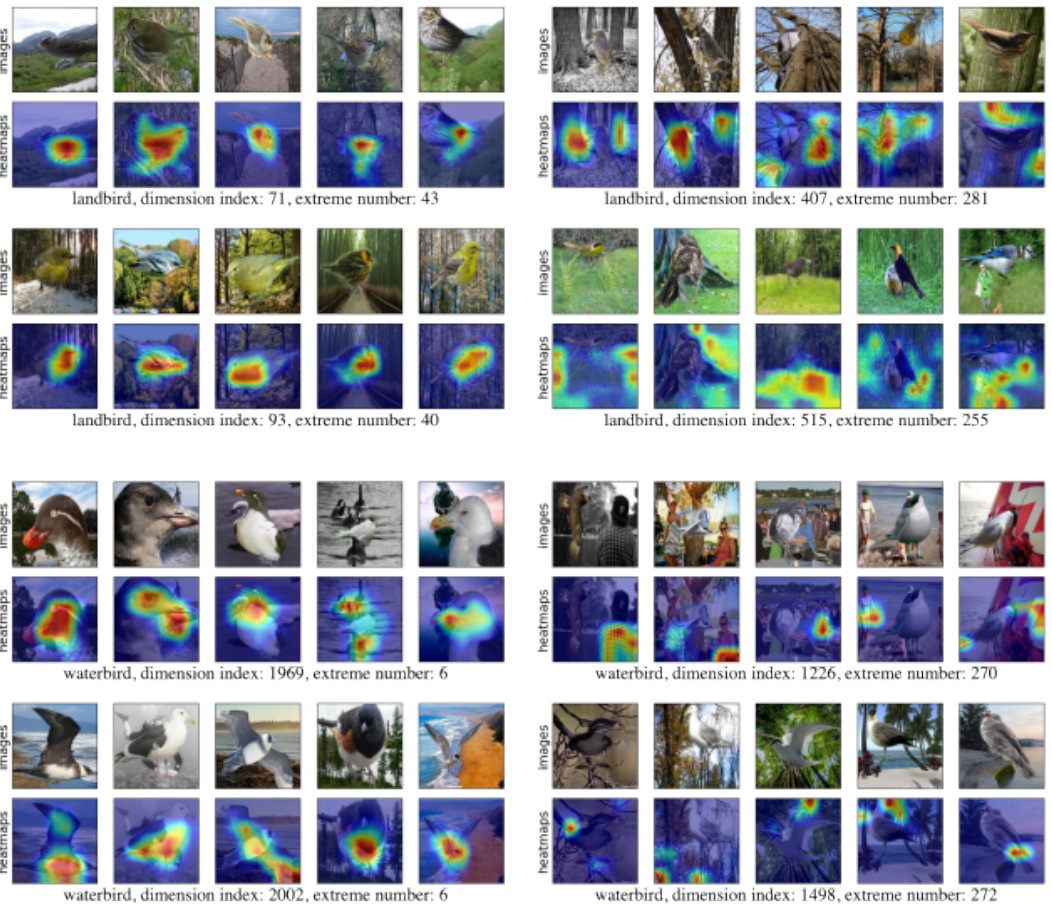

Figure 8: Visualization of selected dimensions from the latent embedding space (2048 dimensions) of a ResNet-50 trained on the Waterbirds dataset. The right side of the figure shows dimensions having many extreme values, while the left side of the figure shows dimensions having few extreme values.

birds dataset. In practice, we select $\beta$ based on how much average performance (which is always available regardless of the group labels) we expect to trade for better worst-group performance. We choose $\beta = 5$ in our experiments as this value selects models that have average performance comparable to baseline methods.

## A.5 RESULTS ON VISION TRANSFORMERS

We aim to understand how our method generalizes to different model architectures. Vision transformers (ViT) (Dosovitskiy et al.) are attention-based neural networks and have a very different mechanism to process images. Different from convolutional networks, ViT models expect the input to be a sequence of image patch embeddings.

We adopted a small version of ViT named ViT-S (Ghosal et al., 2022), which has approximately 21.8M parameters and is pre-trained on the ImageNet-21k (Deng et al., 2009) dataset. We first trained a ViT-S model on Waterbirds and CelebA datasets with ERM, respectively. Then, we extracted the embedding of `class` token from the model for each training sample as its latent representation. Finally, we applied RetroTune to fine-tune the classification layer of the model. We used 224×224 images as inputs. During ERM training, we used an SGD optimizer with an initial learning rate of 0.03, a momentum of 0.9, a weight decay of $10^{-4}$. We set the batch size to 256 and trained the models for 100 epochs. The learning rate decay is controlled by a cosine annealing learning rate scheduler. During RetroTune fine-tuning, we used a constant learning rate of $10^{-5}$,

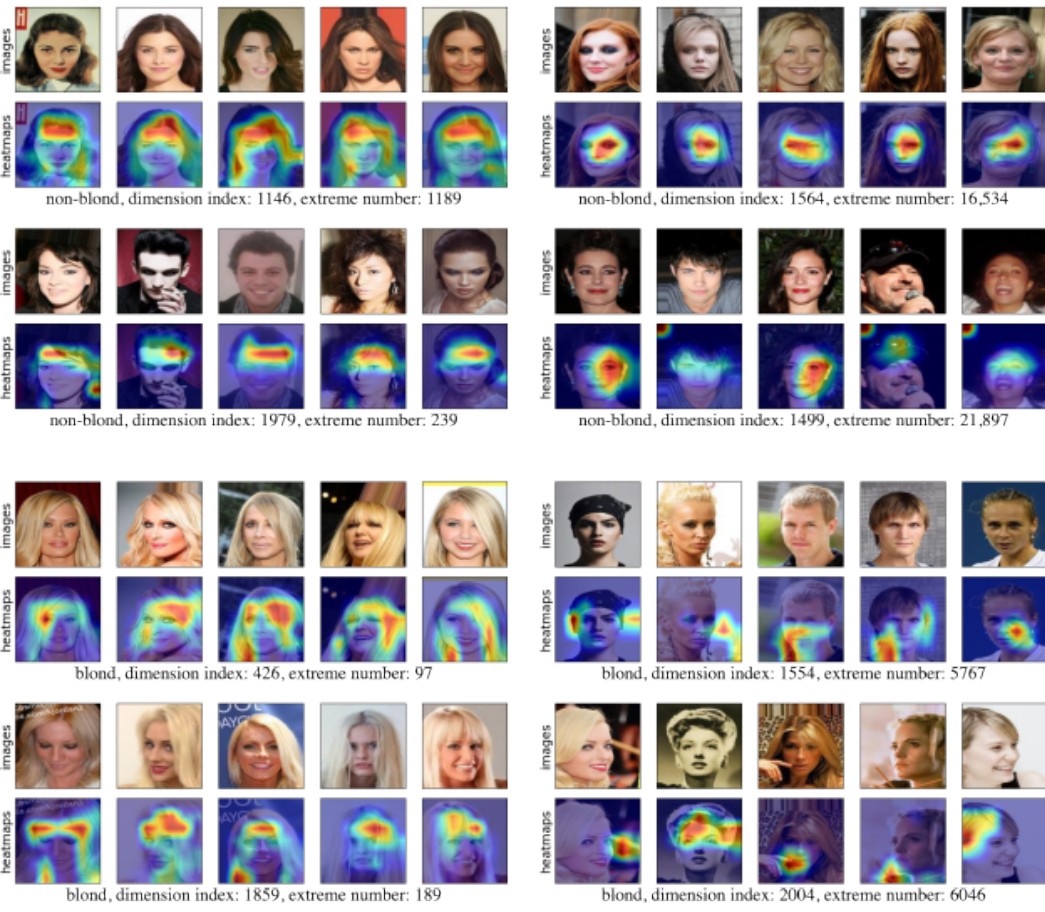

Figure 9: Visualization of selected dimensions from the latent embedding space (2048 dimensions) of a ResNet-50 trained on the CelebA dataset. The right side of the figure shows dimensions having many extreme values, while the left side of the figure shows dimensions having few extreme values.

$\lambda = 0.1$, a batch size of 256 for learning on the Waterbirds dataset; and we used a constant learning rate of $10^{-3}$, $\lambda = 0.1$, a batch size of 256 for learning on the CelebA dataset.

We observe from Table 6 that for a completely different model architecture, RetroTune is still effective, especially on the CelebA dataset where RetroTune improves the worst-group accuracy of the ERM model by 44.95%.

| Method | Waterbirds | | CelebA | |
|---|---|---|---|---|
| | Avg. | Worst | Avg. | Worst |
| ERM | **93.87** | 73.52 | **95.98** | 43.33 |
| RetroTune (ours) | 90.71 | **78.50** | 90.08 | **88.28** |

Table 6: Average and worst-group accuries (%) obtained on the ViT-S models.

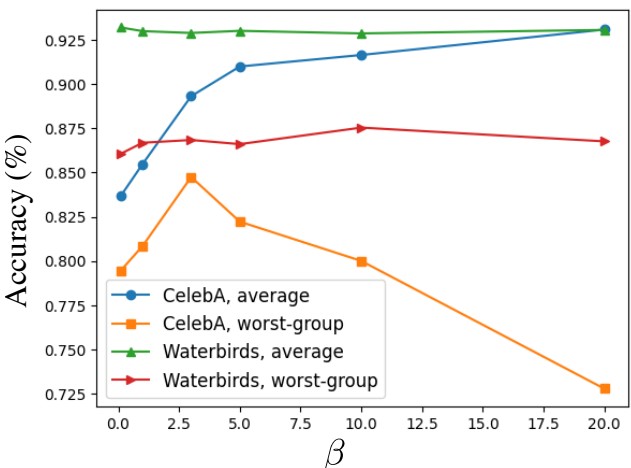

Figure 10: Average and worst-group accuracies (%) of the RetroTune models selected by $\beta$.

