# OpenReview forum: "RetroTune: Mitigating spurious features via retrospective fine-tuning"
_ICLR.cc/2024/Conference — ICLR 2024 Conference Withdrawn Submission_

### Official Review · Reviewer_nfXr · 2023-10-26

**Soundness:** 2 fair
**Presentation:** 3 good
**Contribution:** 2 fair
**Rating:** 3
**Confidence:** 4

**Summary:**

The paper introduces RetroTune, a fine-tuning technique aimed at reducing the impact of spurious correlations in machine learning models. The method distinguishes between core and spurious dimensions in the latent representation by analyzing the number of outliers in each dimension. It then finetunes the last layer (classifier) of the trained model by randomly swapping or masking spurious or core dimensions. Compared to a few existing methods, including LfF, CVaR, DRO, JTT, DFR, DivDis, and ReBias, the proposed method RetroTune achieved higher worst-group accuracy on Waterbirds, CelebA, and BiasedMNIST when no group labels are available for model training and selection.

**Strengths:**

1. The proposed method does not require additional data or labels, offering an unsupervised approach to the problem.
2. The paper is well-structured and offers a clear explanation of the algorithm and its underlying logic.

**Weaknesses:**

1. **Missed Common Setting:** The paper does not address comparisons with other methods in the widely-accepted scenario where a validation set with group labels is available.

2. **Unclear Model Selection for Comparisons:** While the paper specifies the model selection process for RetroTune, it lacks clarity on whether and how this is done for other methods, making it difficult to fully validate the reported results.

3. **Suboptimal Performance and Context Mismatch:** The paper uses datasets with available group labels but does not offer compelling justification for why it focuses on a no-labels scenario. When there is a validation set with group labels, the reported results are not state-of-the-art according to the reported numbers in the existing literature.

**Questions:**

1. The evaluation is limited without a direct comparison with the many existing methods. A lot of related papers are missing here, see a non-comprehensive list below. Multiple lines of group inference methods are directly comparable to the proposed methods so their results should be added, and robust training methods that require group labels should be properly discussed.
2. In Section 3.4, is $a_{err}$ always zero?
3. The reported dimensions considered as extended spurious seem highly disproportionate across classes. There are in total 2048 dimensions and Table 2 shows that more than 2040 dimensions are considered as extended spurious for one class while only 4 or 20 dimensions are spurious for the other class. Can you provide insight into why this is the case and its effect on different classes?
4. Is there a specific reason for comparing only to ReBias on BiasedMNIST without other methods and not having ReBias for the other two datasets, Waterbirds and CelebA?
5. Have you considered extending your evaluations to text-based models and data?


(Sohoni et al.) No Subclass Left Behind: Fine-Grained Robustness in Coarse-Grained Classification Problems. NeurIPS, 2020.

(Creager et al.) Environment Inference for Invariant Learning. ICML, 2021.

(Zhang et al.) Correct-N-Contrast: a Contrastive Approach for Improving Robustness to Spurious Correlations. ICML, 2022.

(Taghanaki et al.) MaskTune: Mitigating Spurious Correlations by Forcing to Explore. NeurIPS, 2022.

(Nam et al.) Spread Spurious Attribute: Improving Worst-group Accuracy with Spurious Attribute Estimation. ICLR, 2022.

(Deng et al.) Robust Learning with Progressive Data Expansion Against Spurious Correlation. NeurIPS, 2023.

---

### Official Review · Reviewer_uLm7 · 2023-10-31

**Soundness:** 2 fair
**Presentation:** 2 fair
**Contribution:** 2 fair
**Rating:** 3
**Confidence:** 4

**Summary:**

This paper studies mitigating the spurious features during model finetuning. The authors leverage several heuristics to define and design methods to identify the spurious and core features. Then they propose to synthesize fine-tuning data by specifically perturbing at the identified dimensions of latent representations to enforce desired changes in the last layer’s weights. On two synthetic datasets with severe bias, Waterbirds and CelebA, the proposed method RetroTune obtained some improvements.

**Strengths:**

(+) The proposed method seems to be interesting;

(+) The visualizations are illustrative;

**Weaknesses:**

(-) The studied setting seems to be limited;

(-) The whole method is designed by multiple heuristics without any theoretical guarantees. The handcrafted heuristics make the paper hard to follow;

(-) Experiments are limited to synthetic datasets;

**Questions:**

1. The studied setting seems to be limited. The whole paper considers OOD generalization under severe bias, without group labels. The implicit assumption makes the problem much easier to solve, while it’s unknown whether it’s the case for many practical scenarios.



2. Throughout the description of the method, there are too many heuristic designs that disable the method being applied to a general setting:
- How can the values of certain dimension reflect single invariant or spurious features? What if they are entangled together?
- Why “a spurious feature does not exist in all the training samples, and values at the corresponding dimension tend to be out-of-distribution for the inputs that do not contain the spurious feature.”? Even for a single type of background in Waterbirds, there are variations inside that is likely to change the values of the latent features. What does it mean by out-of-distribution for the inputs?
- Why “swapping core (spurious) dimensions should (should not) alter the predictions”?
- Why “masking out spurious dimensions should not alter the predictions; and masking out core dimensions should make the predictions uncertain”?
- Why “the p dimensions are very likely to be spurious” in (6)?
- How the heuristic in Sec. 3.3.2 is exactly incorporated in the whole algorithm?
- The model selection method in Sec. 3.4 is heavily dependent on the severe bias assumption.

Besides, there are no theoretical guarantees for the heuristics;


3. Experiments are limited to synthetic datasets, and only considers setting with severe bias. It’s recommended to evaluate the methods going beyond the synthetic data, such as in Wilds benchmark.

4. Results of some baselines have huge gap than those reported by the original paper. For example, in JTT, the reported worst accuracy in Waterbirds is 86.7%, significantly larger than that in Table 4.


5. Several related works are have not been discussed nor compared in the paper:
- For OOD generalization without group labels, [1] show it’s fundamentally impossible to learn the core features. How could RetroTune prevent the failure case?
- The learning of the spurious features studied in the paper also relies on tendency of optimizer to simple features[2]. There are already multiple methods to preventing the simple feature learning[3,4,]. Can these methods address the issue in the paper?
- [5,6,7] employ new methods to learn richer features, instead of spurious features, which could be severed as a more practical variant of Asgari et al., 2022 cited in the paper. Can [5,6,7] address the issue studied in the paper?
- [8,9] are also the state-of-the-art methods for the same problem studied in the paper. Can they address the issue studied in the paper?


**References**

[1] ZIN: When and How to Learn Invariance Without Environment Partition? NeurIPS 2021.

[2] The Pitfalls of Simplicity Bias in Neural Networks, NeurIPS 2020.

[3] Evading the Simplicity Bias: Training a Diverse Set of Models Discovers Solutions with Superior OOD Generalization, CVPR 2022.

[4] Learning an Invertible Output Mapping Can Mitigate Simplicity Bias in Neural Networks, arXiv 2022.

[5] Rich Feature Construction for the Optimization-Generalization Dilemma, ICML 2022.

[6] Learning useful representations for shifting tasks and distributions, ICML 2023.

[7] Towards Understanding Feature Learning in Out-of-Distribution Generalization, arXiv 2023.

[8] Identifying Spurious Biases Early in Training through the Lens of Simplicity Bias, arXiv 2023.

[9] Robust Learning with Progressive Data Expansion Against Spurious Correlation, arXiv 2023.

---

### Official Review · Reviewer_GfBE · 2023-11-01

**Soundness:** 2 fair
**Presentation:** 2 fair
**Contribution:** 2 fair
**Rating:** 5
**Confidence:** 4

**Summary:**

The paper presents a spurious mitigation method for improving the worst-group accuracy by adjusting weight in the last classification layer. The premise of the paper is that weights in the classification in the extreme layer correspond to spurious correlation and a robust model output should not change if the dimension corresponding to the spurious correlations is shuffled. The authors evaluated the method on the Waterbird and CelebA datasets.

**Strengths:**

* The paper is easy to follow (the method section can be improved tho), and the figures are explanatory.

* The paper addresses an important problem in ML --- mitigating spurious correlations or shortcuts. Most of the works assume the availability of group labels or rely on a human-in-the-loop approach --- in contrast, the method proposed doesn't require group annotation.


* The method is evaluated on the Waterbird and CelebA datasets, and comparison with the baselines demonstrates the method's effectiveness.

* Visualization in Figure 3 confirms the spurious features can be detected by the method.

**Weaknesses:**

* The intuition behind the method or theoretical justification is not clear. The authors have not explained why the weights corresponding to the spurious correlations are larger and don't discuss cases where it may not hold true.

* One of the major issues is using the validation dataset in **Section 3.4** to select/find the model. This enables the model to learn features that work well for validation, but it may not work for OOD distribution. This can be seen from **Table 2**, the number of core features for complex datasets such as CelebA and Waterbird is too low (around 10). The model is likely learning spurious correlations that work well for validation/test datasets but will fail on real-world OOD data.
Ideally, validation/test datasets should not be used during training/model selection.

*
> In contrast, a spurious
feature does not exist in all the training samples,

This is not necessarily true. Spurious correlation can be present in the training examples.

* Section 3.2 is difficult to understand, and it's unclear how the method is implemented. Code is also not provided making the reproducibility difficult.

* It is not clear if the extreme values are calculated based on single example or the entire dataset.

* The regularizaton $R$ in **eqn 3** is added to all transformation in **eqn 2**. It is unclear why reducing the predictive certainty in all four cases is useful.

* It is not clear if the method will work if the model is trained with strong L2/L1 regularization, which is quite common to use in training models.

**Questions:**

* How do you handle conflicts between different classes? For e.g., can a dimension in the latent representation be identified as a core feature for class **C1** but as a spurious correlation for class **C2**?

* In **Table 2**, the number of core features is too low. Do you have any justification for why that is the case?

---

### Official Review · Reviewer_ARMu · 2023-11-06

**Soundness:** 3 good
**Presentation:** 3 good
**Contribution:** 2 fair
**Rating:** 5
**Confidence:** 3

**Summary:**

The authors tackle the problem of spurious correlations during supervised learning. They propose a method to tackle the particularly challenging setting where spurious labels are unknown during both training and validation. Their method involves dividing an ERM model's representation space into "spurious" and "invariant" components, by analyzing the extreme values of the distribution of each component. Then, the spurious and invariant dimensions can be augmented in a re-training procedure to learn a final layer that avoid spurious correlations. The authors evaluate their method on typical benchmarking datasets, finding that they outperform the baselines.

**Strengths:**

- The authors tackle a very challenging problem setup, where spurious features are unknown during both training and validation.
- The fact that extreme values separate spurious from invariant components in the representation is a very interesting empirical finding.
- The proposed method effectively outperforms the baselines.

**Weaknesses:**

1. The primary weakness of the paper is that there are no theoretical justifications for the proposed method. For example, why do spurious components tend to have more extreme values, and under what conditions does this occur? In addition, why is the number of extreme samples used as the selection criteria, over a metric like variance, continuous entropy, or value at risk? Having a simple synthetic analytical example would be a good starting point.

2. The proposed method assumes that the representation space naturally disentangles into separate "spurious" and "invariant" components which are independent. The authors don't really discuss this assumption in the paper, or utilize any strategies to encourage disentanglement during the ERM training. As such, it is unclear under what scenarios (empirically or theoretically) this phenomenon occurs, and how generalizable the proposed method is.

3. In particular, one could imagine the inverted waterbirds problem (studied in Appendix B.1. of [1]), where the background is now invariant, and the bird is spurious. Presumably, RetroTune would fail as it would now identify spurious features as invariant (and vice versa), and thus learn a purely spurious predictor. As such, it is unclear to me what the retrospection step is actually identifying or the underlying assumptions required -- is it assuming that spurious features are simpler?

4. The authors should conduct a few more ablations and sweeps, e.g. showing the impact of varying $\lambda$ in Eq (3), or the "1.5" times the IQR in Eq (5).

5. All of the benchmarked datasets are images. The authors should test on MultiNLI and CivilComments, two popular datasets containing spurious correlations in the NLP domain.

[1] Last Layer Re-Training is Sufficient for Robustness to Spurious Correlations. ICLR 2023.

**Questions:**

Please address the weaknesses above.